# Optimizing oral contraceptive timing: Daytime intake reduces doses and enhances efficacy

Brenda Lyn Gavina[1,2,3], Taeyong Lee[4,5], Olive Cawiding[1,6], Sunhwa Choi[7], Sungwook Choi[8], Soyoung Kim[7]*, Jae Kyoung Kim [1,6,9]*

1 Biomedical Mathematics Group, Pioneer Research Center for Mathematical and Computational Sciences, Institute for Basic Science, Daejeon, Republic of Korea, 2 Maritime Academy of Asia and the Pacific, Bataan, Philippines, 3 Institute of Mathematics, University of the Philippines Diliman, Quezon City, Philippines, 4 Epidemiology and Modelling of Antibiotic Evasion Unit, Institut Pasteur, Université Paris Cité, Paris, France, 5 Infectious Diseases and Anti-infective Resistance Unit, Inserm U1018, CESP, UVSQ, Université Paris-Saclay, Montigny-le-Bretonneux, France, 6 Department of Mathematical Sciences, KAIST, Daejeon, Republic of Korea, 7 Innovation Center for Industrial Mathematics, National Institute for Mathematical Sciences, Seongnam, Republic of Korea, 8 M Fertility Clinic, Seoul, Republic of Korea, 9 Department of Medicine, College of Medicine, Korea University, Seoul, Republic of Korea

* skim@nims.re.kr (SK); jaekkim@kaist.ac.kr (JKK)

## Abstract

Contraception is essential for reproductive health and women's empowerment because it allows informed choices about pregnancy prevention. Oral contraceptives (OCs) are a popular method due to their accessibility and high level of effectiveness in attaining contraception through the suppression of ovulation. However, current OC regimens do not consider circadian hormonal rhythms, which significantly influence hormone secretion and drug metabolism. Accounting for circadian rhythms may further reduce the dosage of current formulations, which pose risks, including an increased likelihood of venous thromboembolism. We addressed this gap by developing a mathematical model that integrates circadian rhythms with contraceptive pharmacokinetics. Our results show that daytime OC dosing reduces the required ethinyl estradiol (EE) dose by about 6% and the required dienogest (DNG) dose by about 52% compared to evening dosing, due to the alignment of EE and DNG concentrations with luteinizing hormone production peaks. We further lowered the EE dose by about 67% using an optimal nonconstant regimen and decreased the number of intake days from 21 to 8. This dual-timescale optimization demonstrates how incorporating circadian rhythms can significantly enhance contraceptive regimens, enabling safer and more effective dosing strategies with broader implications for chronopharmacological interventions.

## Author summary

Oral contraceptives (OCs) are widely used to prevent pregnancy and manage reproductive health conditions like endometriosis and polycystic ovary syndrome.

**Data availability statement:** Model codes can be publicly accessed and downloaded from https://github.com/holdon1221/C2D-Opt.

**Funding:** This work was supported by the Institute for Basic Science (grant no. IBS-R029-C3) and the Basic Science Research Program through the National Research Foundation of Korea (NRF) funded by the Ministry of Education (RS-2025-25397599) to JKK), and the National Institute for Mathematical Sciences (NIMS) grant funded by the Korean government (MSIT) (grant no. NIMS-B25810000 to SK and SC). The funders had no role in study design, data collection and analysis, decision to publish, or preparation of the manuscript.

**Competing interests:** The authors have declared that no competing interests exist.

However, current OCs often use higher hormone doses than necessary, increasing the risk of side effects such as blood clots (venous thromboembolism). Although previous studies have tried reducing doses to make these OCs safer, most have overlooked the natural daily hormonal cycles (circadian rhythms) that influence how our bodies interact with exogenous hormones. In this study, we used mathematical modeling to explore whether the timing of taking OCs could reduce hormone doses while maintaining efficacy to inhibit ovulation securely. We found that taking the OC in the daytime significantly lowers the necessary hormone dose compared to taking it in the evening because it better aligns with the body's natural hormone cycles. Further, by adjusting daily doses rather than using a constant dose each day, we dramatically reduced the total hormone dose and reduced the number of drug-taking days from 21 days to just 8 days per cycle. These findings suggest that considering circadian rhythms when planning contraceptive use can significantly reduce the risks of adverse events from exogenous hormones, without compromising the contraceptive's efficacy, providing an exciting new approach to contraception.

## Introduction

Contraception plays a crucial role in reproductive health, family planning, and gender equality, providing individuals with the ability to prevent unintended pregnancies and make informed reproductive choices. Effective contraception contributes to reducing maternity-related morbidity, improving neonatal health, and empowering women by allowing greater control over their reproductive lives. Hormonal contraceptives are widely used for both pregnancy prevention and the management of reproductive disorders such as endometriosis and polycystic ovary syndrome [1–4].

Among various hormonal contraceptive methods, oral contraceptives (OCs), composed of exogenous estrogen and progesterone, are the most widely used due to their accessibility, reversibility, and high efficacy when taken correctly. These contraceptives work by inhibiting ovulation through the suppression of pituitary hormones, particularly follicle-stimulating hormone (FSH) and luteinizing hormone (LH) [5–8]. When adequate follicular development is prevented, estrogen ($E_2$) remains insufficient to trigger the LH surge, thereby blocking ovulation. As a result, progesterone ($P_4$) levels remain below the contraceptive threshold of 3 ng/mL, ensuring effective contraception [5] (Fig 1A).

To address clinical needs, OCs are commonly formulated with a constant daily dose, taken at the same time each day for 21 consecutive days, followed by a 7-day drug break (Fig 1B, left). Fixed-time dosing helps maintain stable hormone levels and maximize efficacy [9,10]. It also supports adherence by simplifying the daily routine and provides a practical strategy to broadly accommodate variability in menstrual cycle onset across individuals and cycles. However, there is no formal guideline specifying whether daytime or evening dosing is preferable, a potential gap in clinical practice. Furthermore, current OC formulations pose challenges such as nausea

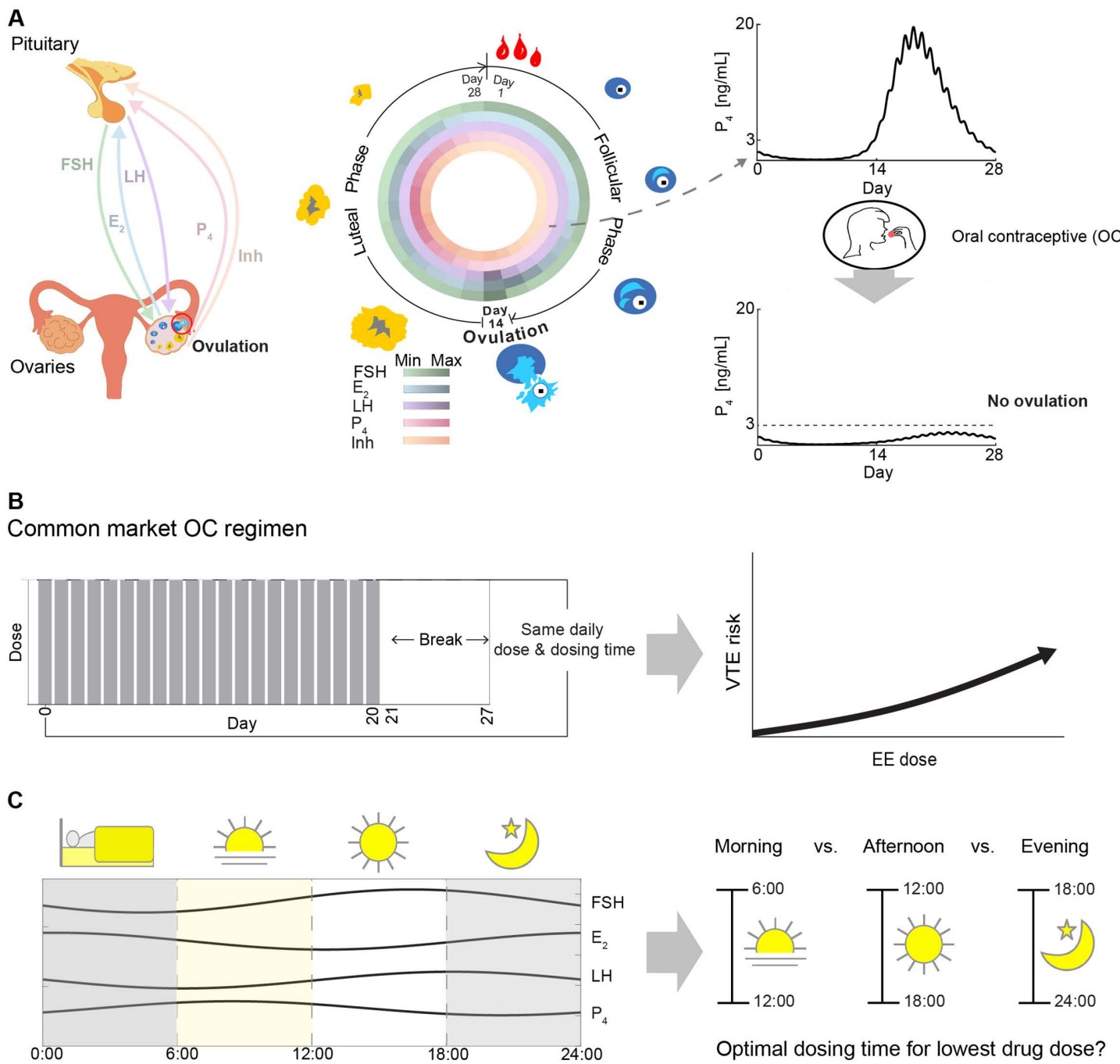

**Fig 1. Research aim of identifying the optimal dosing time for oral contraceptives (OCs). (A)** Follicle-stimulating hormone (FSH), luteinizing hormone (LH), estrogen ($E_2$), progesterone ($P_4$), and Inhibin (Inh) regulate the menstrual cycle to cause ovulation (left panel). Ovulation, the release of an egg on day 14 of a typical 28-day cycle, divides the cycle into the follicular and luteal phases. The follicular phase, driven by FSH and LH, leads to an $E_2$ peak, LH surge, and ovulation. The luteal phase follows with corpus luteum formation and peak Inh and $P_4$ production (middle panel). $P_4$ hormone levels change from normal to anovulatory levels upon administration of OCs (right panel). **(B)** OCs on the market are typically administered with the same dose and dosing time for 21 consecutive days followed by a 7-day break (left panel). However, these doses often cause adverse side effects, with a larger ethinyl estradiol (EE) component increasing venous thromboembolism (VTE) risk (right panel). **(C)** To minimize side effects, the study examines the impact of circadian rhythms of FSH, $E_2$, LH, and $P_4$ on OC dosing (left panel). The aim is to identify the best dosing time —morning, afternoon, or evening—to suppress ovulation with the lowest effective dose (right panel).

and increased risk of venous thromboembolism (VTE), particularly due to greater estrogen content [11–14] (Fig 1B, right). To minimize these risks, numerous studies have explored reducing contraceptive doses [15–18]. In particular, the recent study by Gavina et al. [18] employed mathematical modeling and optimization to identify the minimal effective doses and optimal timing for hormonal contraceptive administration within the menstrual cycle.

However, most recent studies have overlooked the impact of hormonal circadian rhythms on contraceptive efficacy, with the only exception being a study conducted four decades ago [19]. This gap is particularly notable given that circadian rhythms regulate numerous physiological processes, including hormone secretion, metabolism, and drug absorption. Since key reproductive hormones—FSH, LH, $E_2$, and $P_4$—exhibit well-documented circadian variations [20] (Fig 1C, left), it is plausible that the timing of OC intake could influence its efficacy. Despite this, current formulations do not account for daily hormonal fluctuations, raising the question of whether optimizing the dosing time could enhance contraceptive effectiveness while minimizing side effects.

Chronotherapy—the practice of adjusting medication timing to align with circadian rhythms—has been widely studied in other therapeutic areas [21–25]. Numerous studies have demonstrated that adjusting the timing of medication administration can significantly influence drug efficacy and safety [26–37]. For instance, Bowles et al. [38] found that evening dosing of antihypertensive medications led to greater improvements in blood pressure control than daytime dosing. To et al. [39] further observed that administering rheumatoid arthritis medication at bedtime resulted in greater symptom relief compared to conventional dosing schedules. Moreover, substantial evidence supports the influence of dosing time on the efficacy of diabetes treatments [28]. Beyond chronic disease management, circadian timing has also been shown to influence the effectiveness of cancer treatments [29,31,33–37]. For instance, in female patients with diffuse large B-cell lymphoma, afternoon chemotherapy administration was associated with higher survival rates compared to morning treatment [29]. These findings highlight how circadian rhythms modulate physiological responses to medication, underscoring the need to investigate whether similar principles apply to hormonal contraceptives.

To investigate the impact of circadian rhythm on OC formulation, we extended the mathematical model by Gavina et al. [18] by integrating circadian hormonal rhythms and drug pharmacokinetics (PK). The model captures the circadian patterns of LH, FSH, $E_2$, and $P_4$ alongside the PK of the exogenous estrogen, ethinyl estradiol (EE), and the exogenous progesterone, dienogest (DNG), components of the OC under consideration, accurately reproducing hormonal fluctuations observed in normally cycling women and aligning well with experimental drug concentration data. Using this model, we investigated the impact of the dosing time on contraceptive efficacy. Under a constant dosing regimen (i.e., constant daily dose for 21 days followed by a 7-day break), daytime dosing required significantly lower EE and DNG to suppress ovulation than evening dosing. The total EE and DNG doses needed for ovulation suppression were reduced by approximately 6% and 52%, respectively, primarily due to alignment between EE and DNG concentration peaks and LH production peaks. The synchrony of peaks enhances LH levels and inhibits early follicular development, effectively lowering $P_4$ levels. Further analysis of nonconstant dosing regimens (i.e., varying daily doses over 21 days) reinforced the advantage of daytime administration. This approach reduced the total required drug dose by about 67% and shortened the number of intake days to 8, compared to constant dosing with 21 intake days.

Notably, by optimizing both the daily dosing time and the number of days within the 28-day cycle, we achieved effective ovulation suppression with an exceptionally low total hormone dose. This dual-timescale optimization provides a framework for developing contraceptive regimens with enhanced safety while ensuring efficacy, and may be adapted for other circadian-guided treatments.

## Results

### The new model of hormonal contraception accounts for circadian rhythms and drug pharmacokinetics (PK)

We have integrated the observed circadian rhythms of follicle-stimulating hormone (FSH), luteinizing hormone (LH), estrogen ($E_2$), and progesterone ($P_4$) into the baseline model [18] (Fig 2A). Specifically, we first fitted normalized experimental

hormonal circadian data with cosine curves (Fig 2B(i)). These data (red dots in Fig 2B(i)) represent the mean hormone levels of nine normally cycling women during the follicular phase [20]. Next, the fitted daily cosine curves were combined with each day of the simulated 28-day hormone levels from the baseline model (Fig 2B(ii)). The resulting hormone levels with circadian rhythms (black curve in Fig 2B(iii)), serve as targets for calibrating the new mathematical model (see Methods for details).

Then, to simulate using the new model, we utilized as initial conditions the hormone levels at day 0, which represents the first day of menstrual bleeding and marks the start of the menstrual cycle (see Table B in S1 Text for details). The new model successfully reproduces the daily variations of FSH, LH, $E_2$, and $P_4$ (red curve in Fig 2B(iii)), aligning with the target hormones. Additionally, it captures the 28-day cyclical changes of these hormones.

Next, to investigate the impact of circadian rhythms on the pharmacokinetic dynamics of exogenous hormones, we incorporated the PK of the drug into the baseline model, providing a more precise representation of the drug effects.

For the incorporation of drug PK, we first extracted time series data of EE and DNG drug concentrations (Fig 2C(i)). The data comprise mean drug concentrations in normally cycling women over a 24-hour period following the seventh day (day 6 in model simulations) of daily dosing with the OC containing 30 $\mu$g EE and 2000 $\mu$g DNG [40]. We found that the PK of the EE and DNG components of the OC can be described by a two-compartment model (Fig 2C(ii)). Thus, we incorporated this drug model into the baseline model. It was subsequently calibrated to the extracted drug concentration data, yielding simulated EE and DNG concentrations that matched these data (Fig 2C(iii)). Then, we added a fraction of the drug concentrations to endogenous hormones to integrate drug pharmacodynamics (see Methods for details).

**The efficacy of oral contraceptive (OC) in constant administration depends on dosing time**

**Effects of dosing time on anovulation outcomes.** By using the model with circadian rhythms and PK dynamics, we investigated whether the contraceptive efficacy of EE and DNG drugs changes depending on their dosing time of day. To explore this, we simulated a standard market OC regimen with a constant dose administered for 21 consecutive days, followed by a 7-day drug break. This regimen was applied to the previously calibrated normal cycle, and ovulation suppression was evaluated without any further parameter estimation.

Indeed, anovulatory outcomes varied depending on the dosing time. Specifically, the dosing regimen with 25 $\mu$g EE and 1000 $\mu$g DNG (Fig 3A) resulted in different anovulatory outcomes depending on the dosing time. When the OC was administered at 11:00 (Fig 3B, left) the maximum $P_4$ level remained below the contraceptive threshold of 3 ng/mL (black dashed line in Fig 3B) throughout all three cycles, indicating sustained anovulation. However, when administered at 22:00 (Fig 3B, right), the maximum $P_4$ level exceeded the threshold from the second cycle, indicating a loss of contraceptive efficacy.

Next, we further compared the efficacy of dosing between 11:00 and 22:00 by varying dose combinations ranging from 0 $\mu$g to the market doses (30 $\mu$g EE and 2000 $\mu$g DNG, Fig 3A). Step sizes of 0.6 $\mu$g for EE and 40 $\mu$g for DNG, both below the lowest known human-use formulations (5 $\mu$g for EE [41] and 500 $\mu$g for DNG [42]), were chosen to evenly divide the maximum EE and DNG doses into 50 intervals. Then, the maximum $P_4$ levels over ten cycles of treatment (i.e., 280 days) are described in the heat maps (Fig 3C). In the heat maps, the colored regions indicate areas where the maximum $P_4$ level remains below 3 ng/mL, corresponding to an anovulatory outcome. The colored region is broader when the OCs are administered at 11:00, suggesting that a wider range of drug combinations can achieve contraceptive efficacy at that time compared to administration at 22:00.

We then investigated all possible dosing times from 1:00–24:00. For each dosing time, we calculated the lowest total combined EE and DNG doses for one cycle of treatment leading to anovulation (Fig 3D). Overall, daytime dosing required substantially lower hormone levels than evening dosing. Specifically, EE and DNG doses decreased by 6% and 52%, respectively, with daytime dosing (592.2 $\mu$g EE and 6720 $\mu$g DNG) compared to evening dosing (630 $\mu$g EE and 14280 $\mu$g DNG).

PLOS Computational Biology

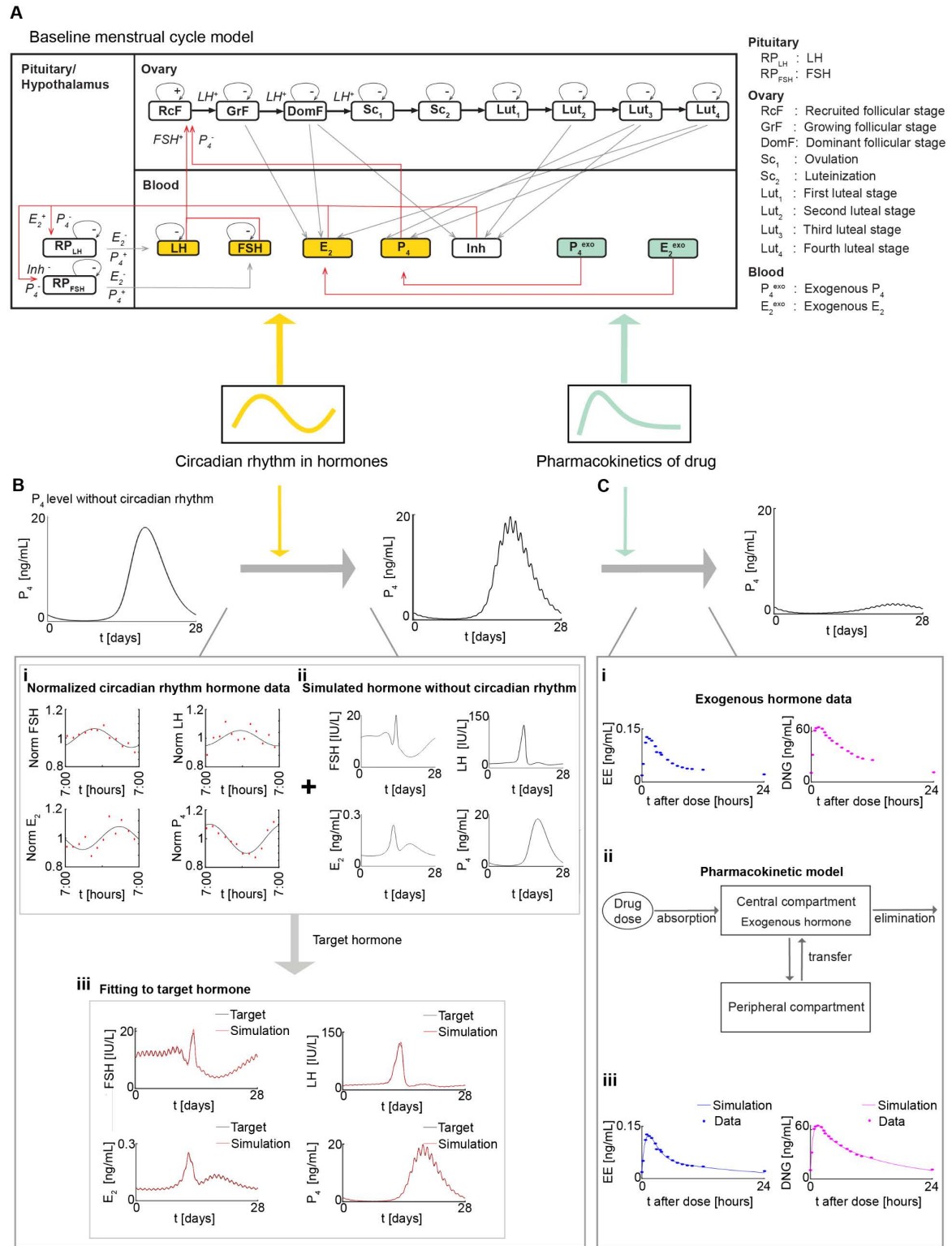

**Fig 2. Development of the new menstrual cycle mathematical model with circadian rhythms and drug pharmacokinetics (PK). (A)** Circadian rhythms are incorporated into LH, FSH, $E_2$, and $P_4$ components (yellow) of the baseline menstrual cycle model. The baseline model is further modified with the inclusion of the PK of drugs (green) in the administration of exogenous hormones $P_4^{exo}$ and $E_2^{exo}$ (see S1 Text for a detailed description of the

model) **(B)** To incorporate the circadian rhythms, experimentally measured daily circadian rhythms (B(i)) are combined to the simulated hormone levels without circadian rhythms (B(ii)) for each day over the 28-day period. The resulting hormone levels with circadian rhythms are used to calibrate the model (B(iii)). **(C)** To incorporate the PK of drugs, extracted ethinyl estradiol (EE) and dienogest (DNG) drug concentration data (C(i)) are used to fit the two-compartment model (C(ii)) to obtain the simulated drug concentration curves (C(iii)).

**Robustness of dosing-time effects under parameter perturbations.** To evaluate whether the advantage of daytime dosing holds under biological variations, we conducted a series of sensitivity analyses. These included perturbations in PK parameters, phase shifts in reproductive hormones, and changes in the underlying menstrual cycle length.

We first perturbed by $\pm 30\%$ the model's 10 non-baseline pharmacokinetic parameters, comprising the volume of distribution ($V_c$) and the absorption ($k_a$), transfer ($k_{21}$), distribution ($\alpha_1$), and elimination ($\beta_1$) rate constants for both EE and DNG. For each perturbation, we calculated the minimum total EE and DNG doses required to induce anovulation for each dosing time. Across all cases, daytime dosing between 10:00 and 12:00 consistently remained optimal, as it corresponded to the lowest OC doses (see Fig C in S1 Text).

OCs on the market encompass a wide variety of PK profiles, including both short- and long-acting formulations. We selected the drug with a short half-life progestin, which makes it more susceptible to circadian variation and may be more sensitive to dosing time. To generalize our findings beyond this specific PK profile, we conducted additional simulations by reducing the DNG elimination rate to one-half and one-fourth of its unperturbed value, mimicking 2× (~19 hours) and 4× (~38 hours) increases in half-life ($t_{1/2,DNG} \approx 9.5$ hours). These extended half-lives are well above DNG's reported clinical range of 7.5–12.2 hours [43,44]. We then applied constant EE and DNG dosing to evaluate the dose required for anovulation across all dosing times. Although the optimal (7:00–12:00) and least effective (18:00–24:00) dosing windows remained unchanged (see Fig D in S1 Text), longer half-lives reduced both total dose and circadian sensitivity. In particular, 2× half-life lowered the DNG optimal dose and reduced the daytime-evening dose gap to 50% (2520 $\mu$g vs. 5040 $\mu$g) (see Fig D in S1 Text), compared to 53% (6720 $\mu$g vs. 14280 $\mu$g) in the unperturbed case (see Fig 3D). With a 4× longer half-life, this difference further narrowed to 33% (1680 $\mu$g vs. 2520 $\mu$g), supporting the conclusion that longer half-life drugs are less sensitive to circadian variation.

Next, we evaluated the effect of shifting baseline hormone curves. This accounts for the fact that the Welt dataset, which was used to fit these curves, has only daily measurements, making the true timing of hormone fluctuations within each day uncertain and could plausibly shift by up to $\pm 12$ hours. After applying these $\pm 12$-hour shifts, we reintroduced circadian rhythm, re-estimated circadian parameters, and simulated constant dosing across various EE and DNG dose combinations. For all cases, daytime administration (7:00–12:00) again emerged as the most effective for ovulation suppression (see Figure E in S1 Text). Interestingly, lower total combined doses of EE and DNG are required to induce anovulation in the -12-hour baseline shift case compared to +12-hour baseline shift case. Its underlying mechanism would be an interesting direction for future work.

Finally, we examined whether our findings hold across menstrual cycle lengths spanning the physiologically typical range, which varies considerably both within and between individuals [45]. To do this, we first generated different cycle lengths by following the approach of Hendrix et al. [46] and Gavina et al. [18] of adjusting model parameters. Specifically, we scaled parameters related to hormone regulation and follicular development ($V_{0,LH}$, $V_{1,LH}$, $V_{FSH}$, $k_{LH}$, $k_{FSH}$, $\alpha_{LH}$, $\alpha_{FSH}$, $b$, $c_1$–$c_4$, $d_1$, $d_2$, and $k_1$–$k_4$) from 1.0985 to 0.9333 to simulate cycles ranging from 25-30 days, which reflects the typical range observed in healthy individuals [47]. We then applied constant EE and DNG dosing to evaluate the dose required to induce anovulation. Regardless of cycle length, daytime dosing consistently outperformed evening administration (see Figure F in S1 Text).

Taken together, while parameter perturbations altered the minimum EE and DNG doses for ovulation suppression, our key finding remained consistent: daytime dosing is more effective than evening dosing across all tested scenarios.

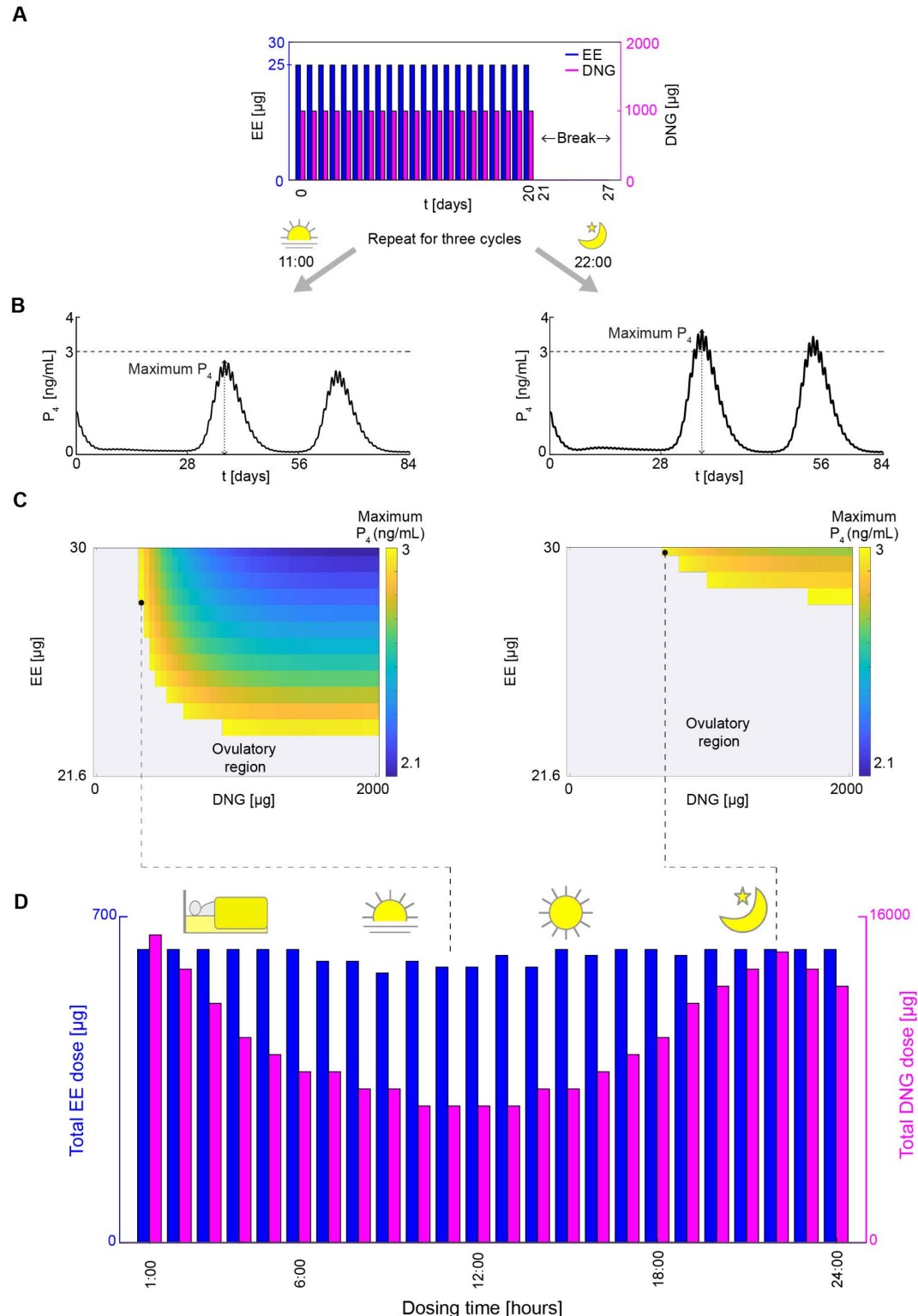

**Fig 3. Lowest total constant dose inducing anovulation for each dosing time. (A)** A dosing regimen of 25 $\mu$g EE (blue bars) and 1000 $\mu$g DNG (magenta bars) for 21 consecutive days, followed by a 7-day drug break, is administered over three cycles at 11:00 (daytime) and 22:00 (evening). **(B)** The resulting maximum $P_4$ level for daytime (left panel) is lower compared to evening (right panel) dosing. **(C)** Color scale represents maximum level

when anovulation is achieved. Gray regions represent the cases when maximum level is higher than 3 ng/mL (i.e., ovulation). The maximum levels for each EE and DNG dose combination, ranging from 0 $\mu$g to the market doses 30 $\mu$g EE and 2000 $\mu$g DNG, are displayed in the heat map for daytime (left panel) and evening (right panel) dosing. For the wider range of dosing combinations, daytime dosing leads to anovulation more efficiently than the evening dosing. **(D)** The lowest total EE and total DNG doses are calculated for dosing times from 1:00–24:00. The lowest total EE (blue bars) and total DNG (magenta bars) doses for 21 consecutive days leading to anovulation are lower with daytime dosing than evening dosing. Specifically, the 11:00 dosing requires a minimum total EE dose of 592.2 $\mu$g (28.2 $\mu$g 21 days) and a minimum total DNG dose of 6720 $\mu$g (320 $\mu$g 21 days). In contrast, the 22:00 requires a minimum total EE dose of 630 $\mu$g (30 $\mu$g 21 days) and a minimum total DNG dose of 14280 $\mu$g (680 $\mu$g 21 days).

**Variation in contraceptive effect is driven by the synchrony between drug concentration and circadian rhythms.** To understand the variation in dose requirements across different dosing times, we first investigated which hormonal circadian rhythms are critical for the variation in the dosing effect. To this end, we conducted additional simulations in which the circadian rhythm of one hormone was removed at a time. Our analysis revealed that the circadian rhythms of the luteinizing hormone (LH) are the primary driver among the four hormones— LH, follicle-stimulating hormone (FSH), estrogen ($E_2$), and progesterone ($P_4$) as it produced the largest difference in maximum $P_4$ levels between daytime and evening dosing (Figure G in S1 Text). Based on this finding, we performed follow-up simulations retaining the circadian rhythms of LH in the model while removing those of the other hormones to simplify the subsequent analysis.

Using the simplified model, we simulated the optimal constant regimen of 28.2 $\mu$g EE and 320 $\mu$g DNG (Fig 3) at two different dosing times: daytime (11:00) and evening (22:00).

When the drug was administered in the daytime (11:00), the EE and DNG concentration peak (Fig 4A) closely aligned with the circadian peak and the increasing phase of pituitary LH ($RP_{LH}$) production (Fig 4B). In contrast, when the drug was administered in the evening 22:00), the EE and DNG concentration peak (Fig 4C) misaligned and occurred during the decreasing phase of $RP_{LH}$ production following its peak (Fig 4D). This difference in alignment resulted in better synchronization between drug absorption and the circadian rhythm of $RP_{LH}$ production for daytime dosing compared to evening dosing. As a result, estrogen ($E_2$), composed of endogenous estrogen and EE concentration, exhibited greater synchrony with the circadian rhythm of $RP_{LH}$ production in the daytime. This synchronization amplified the production term of $RP_{LH}$, leading to higher $RP_{LH}$ levels (Fig 4E) and, consequently, elevated LH levels (see Eqs 1 and 2 in S1 Text for details). Higher LH levels (Fig 4F) accelerated the transition from recruited follicular (RcF) to growing follicular (GrF) stage mass (see Eq 5 in S1 Text for details). This accelerated transition, combined with follicle growth inhibition from DNG daytime dosing, further reduced RcF growth and weakened RcF feedback (Fig 4G). This reduction in RcF initiated a cascade effect, sequentially decreasing follicular and luteal masses, from GrF to the fourth luteal stage mass ($Lut_4$) (see Eqs 6–13 in S1 Text for details). Finally, the reduced third luteal stage mass ($Lut_3$) and $Lut_4$ contributed to lower progesterone ($P_4$) levels with daytime dosing (yellow curves in Fig 4H–4J) compared to evening dosing (black curves in Fig 4H–4J).

## The efficacy of oral contraceptive (OC) in nonconstant administration depends on dosing time

The lower $P_4$ levels with daytime (at 11:00) compared to evening dosing (at 22:00) under the same constant dose (Fig 4J) indicate that daytime dosing is more efficient in terms of reducing $P_4$ levels to anovulatory levels. For example, Fig 3D and Fig 5A demonstrate that the optimal constant regimen requires lower EE and DNG doses for daytime dosing at 11:00 (28.2 $\mu$g EE, 320 $\mu$g DNG) compared to evening dosing at 22:00 (30 $\mu$g EE and 680 $\mu$g DNG). This suggests that less drug is needed in the daytime to suppress ovulation effectively.

To further reduce the dose requirement, we allowed the drug dose to vary over the first 21 days of the 28-day treatment cycle instead of maintaining a constant level. Specifically, the EE dose for each of these 21 days can be varied from 0 $\mu$g to the market dose of 30 $\mu$g, while the DNG dose can be varied from 0 $\mu$g to the market dose of 2000 $\mu$g. Next, we applied optimization to minimize the total EE and DNG doses while maintaining anovulation across ten cycles (see Methods for details). This resulted in the optimal nonconstant regimen for each dosing time.

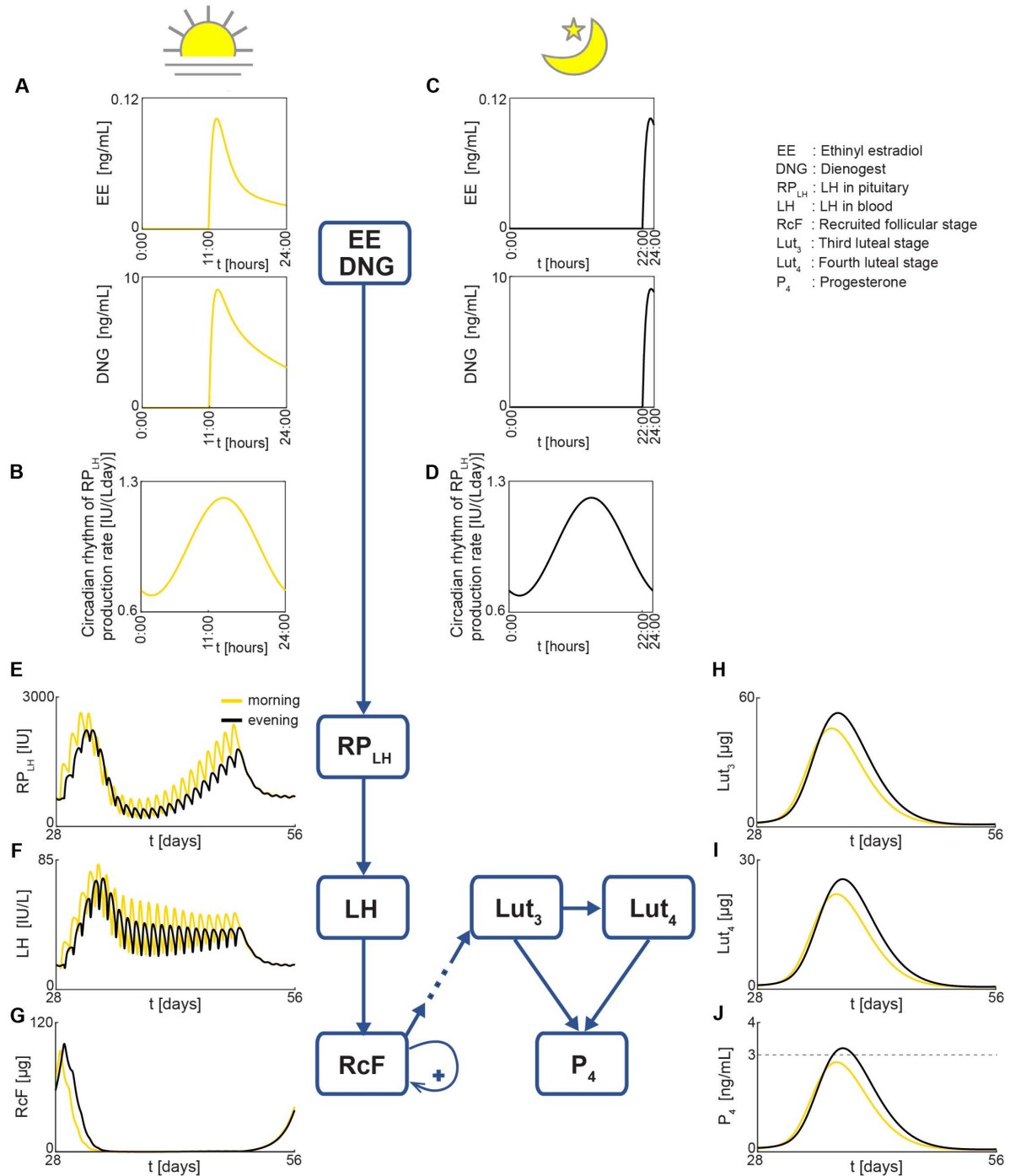

**Fig 4. Impact of drug concentration peak alignment with circadian rhythms on P₄ levels.** **(A–D)** The EE and DNG dose concentration peak for daytime dosing (A) nearly aligns with the circadian rhythm of the $RP_{LH}$ production rate (B), whereas evening dosing (C) results in an EE and DNG peak after the peak of the $RP_{LH}$ production rate, causing misalignment (D). **(E–F)** Alignment between daytime dosing and $RP_{LH}$ production rate increases $RP_{LH}$ levels (E), leading to a higher LH (F). **(G)** Higher LH further accelerates the RcF transfer to the next follicular stage, while DNG from daytime dosing inhibits RcF development, lowering the early RcF peak, weakening RcF feedback, and thereby reducing RcF masses more than with evening dosing. **(H–J)** Consequently, follicular and luteal masses decrease more in subsequent stages for daytime dosing, leading to lower $Lut_3$ (H) and $Lut_4$ (I), and ultimately lower $P_4$ levels (J) compared to evening dosing.

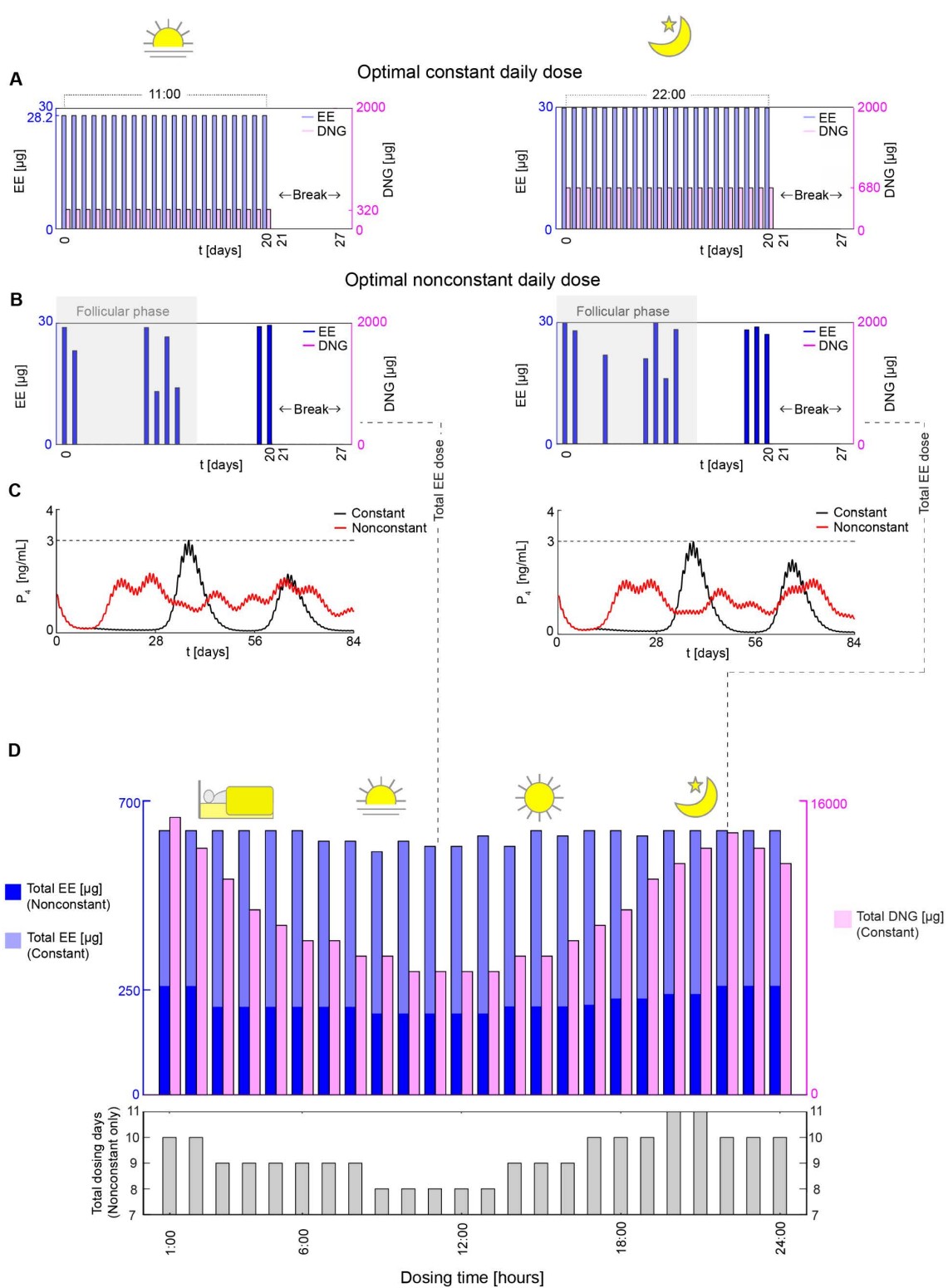

**Fig 5. Lowest total nonconstant dose inducing anovulation for each dosing time. (A)** Anovulation is achieved with an optimal constant daily dose of 28.2 µg EE and 320 µg DNG administered at 11:00 in the daytime (left panel), while the optimal constant regimen at 22:00 in the evening requires 30 µg EE and 680 µg DNG (right panel). **(B)** When the daily dose is nonconstant, the optimal regimen has a significantly lower total EE dose than constant

dosing. The optimal nonconstant daytime regimen (left panel) has a lower total EE dose (193.78 $\mu$g) and fewer intake days, 8 days, than the optimal nonconstant evening regimen (right panel), which has a total EE dose of 259.88 $\mu$g and 10 intake days. Both the daytime and evening regimens have no DNG dose for nonconstant dosing, unlike constant dosing (light magenta bars). Additionally, the nonconstant regimens exhibit substantial dosing during the follicular phase (gray shaded region). **(C)** When the optimal nonconstant regimen is repeated three times, the $P_4$ levels (red) show no $P_4$ relapse, whereas constant dosing (black) presents a relapse in the second cycle (left panel). **(D)** Consequently, the optimal total EE dose (blue bars) for nonconstant dosing is lower than the optimal total EE dose (light blue bars) for constant dosing. Further,the total nonconstant EE dose at 11:00 and the rest of the daytime dosing is lower than for nonconstant evening dosing, with fewer intake days, 8–9 days versus 10–11 days.

The optimal nonconstant regimens at both 11:00 (Fig 5B, left) and 22:00 (Fig 5B, right) exhibit substantial dosing during the follicular phase (gray shaded region). These regimens resulted in a maximum $P_4$ concentration over three cycles (red curves in Fig 5C) below the contraceptive threshold of 3 ng/mL (black dashed line). However, the evening dosing regimen (22:00) required 29% more EE (205 $\mu$g) than the daytime regimen (10:00) (158 $\mu$g).

Subsequently, we compared the total drug doses required for the optimal nonconstant regimen across all dosing times (blue bars in Fig 5D). Notably, evening dosing required larger total doses than daytime dosing. In particular, the worst dosing times (22:00–24:00) required 34% more EE (259.88 $\mu$g) than the best dosing times (9:00–13:00) (193.78 $\mu$g). Additionally, we observed variations in the total number of dosing days. daytime dosing required fewer intake days (8–9) compared to evening dosing (10–11) (gray bars in Fig 5D). This illustrates the efficiency of daytime administration in reducing overall drug exposure compared to evening dosing.

Furthermore, nonconstant dosing required a significantly lower total dose (193.78 $\mu$g), about 67% less compared to constant dosing (592.2 $\mu$g). We found that such reduction in dose level stems from the unique pattern of the optimal nonconstant regimen with drug-free periods (Fig 5B). These dosing gaps led to a significantly lower intake frequency (8–11 days) compared to the 21 days required for constant dosing. As a result, the nonconstant regimen can effectively keep $P_4$ below the anovulatory threshold throughout the cycle (Fig 5C) while using a lower total dose.

## Discussion

By developing a new model of hormonal contraception that integrates circadian rhythmicity and drug PK, we found that daytime is the most effective time for oral OC administration. This is attributable to the close alignment between the EE and DNG concentration peak and the LH production peak. To the best of the researchers' knowledge, no recent studies have investigated the optimal time of day for OC administration. A study by Kiriwat and Fotherby [19] reported no significant difference in the effectiveness of daytime versus evening dosing for the contraceptive formulation containing ethinyl estradiol and norethisterone. While this study provided early insight into time-of-day effects, its conclusions were based on the methodological tools and sample sizes available at the time. With advances in clinical trial design and hormone monitoring, modern and larger-scale studies could offer a more robust assessment of the impact of dosing time on the efficacy of current OC formulations.

Under constant administration, the repeated application of the dosing regimen led to a decline in effectiveness, as evidenced by $P_4$ relapse in the second cycle (Fig 3B). This observation is consistent with the findings of Spona et al. [48], who reported greater follicular activity in the second cycle than in the first treatment cycle. Our simulation suggests that this effect arises because the first treatment cycle alters hormone dynamics in subsequent cycles. Specifically, our results indicate that $P_4$ levels in the second cycle advance earlier in phase than in the first cycle. Consequently, when the same dosing regimen is repeated, the drug administration becomes delayed relative to the earlier rise in $P_4$ levels, reducing its ability to effectively suppress ovulation and leading to $P_4$ relapse.

Interestingly, nonconstant dosing prevents the second cycle $P_4$ relapse by incorporating drug-free intervals in the optimal dosing regimen. As a result, less total drug is needed to maintain anovulation. However, clinical studies suggest that gaps in drug administration may compromise ovulation suppression [49,50]. While most studies report that such gaps can

lead to ovulation, some findings indicate that anovulation may still be maintained depending on the timing of drug administration [51]. In this work, we were able to identify drug-free intervals that maximize efficacy while minimizing drug exposure using an optimization approach. This aligns with our findings, where the optimal regimen exhibits a substantial drug dose administered during the follicular phase (Fig 5B, gray shaded region), highlighting this phase as the most effective timing for ovulation inhibition. Additionally, the regimen begins with a high-dose administration on the first day of the cycle, underscoring the importance of early hormone regulation in suppressing ovulation. This finding concurs with a clinical study by Østergaard and Starup [52], which demonstrated that initiating administration before the 7th day of menstruation enhances the reliability of ovulation suppression.

We also found that ovulation can be suppressed using EE alone, as the optimal nonconstant regimen relies exclusively on EE. However, estrogen-only contraceptives have limited clinical acceptance primarily due to the risk of estrogen breakthrough bleeding and an increased risk of endometrial cancer, while higher estrogen doses are also associated with an increased risk of VTE [13,53]. Consequently, research over the years has prioritized minimizing estrogen dosage while compensating with synthetic progesterone, such as DNG, to maintain ovulation inhibition and to protect the endometrium from the effects of unopposed estrogen. Although our optimal results suggest an EE-only regimen, likely due to our model not accounting for estrogen-associated risk, our simulations demonstrate that supplementing this with DNG can sustain anovulation for ten cycles (Fig H in S1 Text). This observation provides a basis for developing a combined OC regimen with an optimized estrogen dose.

The suggested optimal nonconstant OC regimen aligns with established clinical approaches, as demonstrated by Qlaira, a marketed OC that varies both hormone type and dose across four stages of the cycle [54]. This shows that time-varying regimens are feasible in practice. Moreover, user adherence can be supported by strategies such as color-coded packaging and digital reminders, already widely used in modern drug formulations.

While our study presents a novel perspective, it is important to acknowledge the inherent limitations in our model. First, menstrual cycle length exhibits both intra- and inter-individual variability [45,47,55]. Though our model explored inter-individual variability by simulating different cycle lengths (Fig F in S1 Text), intra-individual variability was not addressed and should be considered in future work to enhance predictive accuracy. Additionally, the model assumes a uniform menstrual cycle onset, overlooking variations across individuals and cycles, limiting its representation of physiological diversity. Furthermore, future investigations of circadian rhythm effects using other models, such as that of Wright et al. [56] and the more complex model by Reinecke and Deuflhard [57,58], would help determine whether the observed optimal dosing time is model-independent. In addition, employing a detailed model could provide a more physiologically realistic estimate of the optimal drug dose for each dosing time. Another limitation arises from our use of only the follicular-phase circadian data to represent the entire menstrual cycle. Although circadian rhythms differ only slightly between the follicular and luteal phases [20], incorporating both would require a piecewise-defined model, substantially increasing computational cost. Future work with faster optimization could incorporate phase-specific circadian dynamics to better capture hormonal regulation. Moreover, the follicular-phase circadian data are limited in both sample size and age range. Although these factors constrain the generalizability of the results, integrating diverse hormone profiles could enhance the model's robustness and applicability across broader populations. Additionally, we modified the relative binding affinity (RBA) values of EE and DNG to simulate their contraceptive effects. This adjustment may be necessary due to the simplifications in our mathematical model, which, while capturing key hormonal and pharmacokinetic dynamics, does not fully represent the drug's pharmacodynamics. Applying our optimization framework to more detailed menstrual cycle models, such as that of Reinecke and Deuflhard [57,59], could provide a more physiologically realistic representation of drug kinetics and interactions.

It is important to note that our model does not currently account for potential side effects associated with the administration of EE, including nausea or other adverse reactions such as VTE. Incorporating these effects would require expanding the model accordingly. Finally, our model, consistent with the frameworks of Gavina et al. [18] and Wright et al. [56], characterizes anovulation not as a complete absence of corpus luteum formation, but as a marked reduction in luteal tissue

mass. This reflects physiologically relevant scenarios such as incomplete luteinization or luteinized unruptured follicle (LUF) [60], where luteal development is impaired but not absent. Importantly, low-amplitude progesterone fluctuations have been observed in such conditions. For instance, progesterone levels in certain LUF cycles were measured at less than 2.5 ng/mL [61], below the anovulatory threshold, supporting the physiological plausibility of our model outputs. While current model aligns with clinically observed variations in ovulatory dysfunction, future model refinements could develop the model to represent anovulation cases in which corpus luteum formation is absent, thereby encompassing a broader range of ovulatory outcomes. Additionally, we evaluated contraceptive efficacy based solely on the primary mechanism of hormonal contraceptives (ovulation inhibition), measured by $P_4$ levels. However, $P_4$ alone may not be a definitive indicator of anovulation, as elevated levels can occasionally be observed even in the absence of ovulation. Nevertheless, since $P_4$ is generally well correlated with ovulatory status, we considered ovulation to be suppressed when $P_4$ levels remained below the established threshold [55]. Future models could include other indicators of anovulation for a more comprehensive evaluation.

In this study, we investigated optimal constant dosing regimens (Fig 3) and nonconstant dosing regimens (Fig 5). In both regimens, daytime dosing was consistently more effective than evening dosing. However, greater caution is needed when translating the nonconstant dosing regimens into clinical practice compared to constant dosing regimens, as the latter are well-established and clinically tested [55], whereas the optimal nonconstant dosing regimen requires further validation in clinical settings. Despite this limitation, our findings highlight the importance of dosing time in optimizing contraceptive efficacy and safety, providing a foundation for future clinical investigation and personalized contraceptive strategies.

## Materials and methods

### Hormonal circadian rhythm and drug concentration data

We used experimentally measured circadian rhythms of luteinizing hormone (LH), follicle-stimulating hormone (FSH), estrogen ($E_2$), and progesterone ($P_4$) (Fig 2B(i)) during the follicular phase of normally cycling women aged 19–29 years. Moreover, these data covered 24 hours post-wake, with wake time defined as 07:00. The data were extracted from a study by Rahman et al. [20] using the software DigitizeIt [62].

Then, to incorporate drug PK into our model, we used experimentally measured drug concentrations of the OC Valette. This drug is selected due to its short half-life, making it particularly susceptible to circadian variations [63]. It consists of EE with a half-life of 6–27 hrs and DNG with a half-life of 7.5-12.2 hrs. The mean EE and DNG concentration data from normally cycling women covered a 24-hour period following the seventh day of daily dosing with the drug. These data were extracted from the Public Assessment Report for Dienogest/Ethinylestradiol Exeltis, published by the Swedish Medical Products Agency [40] using DigitizeIt [62].

### New hormonal contraception model with circadian rhythm and pharmacokinetics

**Model development.** To date, several mathematical models of hormonal contraception have been developed [18,56,57,59,64–69]. Among them, the model by Reinecke and Deuflhard [57,59] offers a highly detailed and physiologically realistic representation but at the cost of considerable complexity, incorporating 49 DDEs and 208 parameters. By comparison, the Wright [56] and Gavina [18] models, both derived from the work of Margolskee et al. [70], are simpler, comprising 17 and 16 DDEs, respectively. Notably, the Wright model includes an equation for the effect of estrogen on progesterone, although this characterization is limited, as acknowledged by the authors. Given these considerations, we adopted the Gavina et al. [18] model as our baseline, as its simplicity was sufficient for investigating dosing-time effects on OCs while facilitating analysis and reducing computational time. All assumptions, definitions, equations, and parameter settings from this model were retained. While this baseline model captures the key hormonal feedback loops of the menstrual cycle and provides a mechanistic basis for evaluating the effects of exogenous estrogen and progesterone, it lacks circadian regulation and drug PK, both essential for evaluating time-of-day dosing effects. By incorporating these features, we developed a model with the circadian rhythms (Fig 2A, yellow) of LH, FSH, $E_2$, and $P_4$, as

well as the PK (Fig 2A, green) of EE and DNG. Detailed descriptions of model modifications are provided in the following section, along with the full model and all parameter values and settings in S1 Text.

**Incorporation of circadian rhythm into the baseline model.** We first incorporated the circadian rhythms of the reproductive hormones LH, FSH, $E_2$, and $P_4$ into the baseline model to reproduce circadian amplitude and phase patterns observed in normally cycling women during the follicular phase [20]. To do this, we adopted cosine functions, which provided a good fit to the target hormone profiles (see Fig 2B(iii)) while offering mathematical simplicity through a small set of parameters—specifically, the amplitudes ($\theta_1$, $\theta_3$, $\theta_5$, $\theta_7$) and phases ($\theta_2$, $\theta_4$, $\theta_6$, $\theta_8$) of the rhythms. This small number of parameters, compared to alternative formalisms such as Fourier series, splines, or mechanistic oscillator-based models, enhances interpretability and has led to their widespread use in chronobiological modeling, including applications in reproductive, cardiovascular, and metabolic systems [71–77]. In addition, having fewer parameters facilitated efficient optimization within our 42-parameter multi-objective framework (see subsection "Optimal regimen for nonconstant drug administration"). For these reasons, we used cosine-based formulations to depict the circadian dynamics of the reproductive hormones.

For LH and FSH—each modeled with separate production, release, and clearance steps (see S1 Text for full model equations)—circadian modulation was applied specifically to the production terms, reflecting the upstream influence of gonadotropin-releasing hormone (GnRH). GnRH is rhythmically secreted by the hypothalamus and acts as a key circadian regulator of LH and FSH biosynthesis [78]. This made the production stage the most suitable point for introducing circadian influence into the model.

Accordingly, the circadian rhythm of LH was implemented by introducing a cosine function into the production term of the pituitary compartment ($RP_{LH}$). Specifically, the LH synthesis term (first term in Eq 1) was modified from its original form

$$\frac{d}{dt}RP_{LH}(t) = \frac{V_{0,LH} + \frac{V_{1,LH}E_2(t)^8}{Km_{LH}^8 + E_2(t)^8}}{1 + P_4(t)/Ki_{LH,P}} - \frac{k_{LH}[1 + c_{LH,P}P_4(t)]RP_{LH}(t)}{1 + c_{LH,E}E_2(t)} \tag{1A}$$

as follows:

$$\frac{d}{dt}RP_{LH}(t) = \frac{V_{0,LH} + \frac{V_{1,LH}E_2(t)^8}{Km_{LH}^8 + E_2(t)^8}}{1 + P_4(t)/Ki_{LH,P}}\left(1 + \theta_1 \cos(2\pi(t - \theta_2))\right) - \frac{k_{LH}[1 + c_{LH,P}P_4(t)]RP_{LH}(t)}{1 + c_{LH,E}E_2(t)}, \tag{1}$$

where $RP_{LH}(t)$ is LH produced in the pituitary awaiting release into the bloodstream, $LH(t)$ is LH blood concentration, $\theta_1$ is the circadian amplitude of LH, and $\theta_2$ is the LH circadian acrophase at time $t$ in days.

Next, we added the follicle-stimulating hormone (FSH) circadian rhythm, attained by incorporating fluctuations into $RP_{FSH}$. For this, the original FSH synthesis term (first term in Eq 3A) in $RP_{FSH}$ was modified from its original form

$$\frac{d}{dt}RP_{FSH}(t) = \frac{V_{FSH}}{1 + Inh(t - \tau)/Ki_{FSH,Inh} + P_4(t)/w} - \frac{k_{FSH}[1 + c_{FSH,P}P_4(t)]RP_{FSH}(t)}{1 + c_{FSH,E}E_2(t)^2} \tag{3A}$$

as follows:

$$\frac{d}{dt}RP_{FSH}(t) = \frac{V_{FSH}}{1 + Inh(t - \tau)/Ki_{FSH,Inh} + P_4(t)/w}\left(1 + \theta_3 \cos(2\pi(t - \theta_4))\right)$$
$$- \frac{k_{FSH}[1 + c_{FSH,P}P_4(t)]RP_{FSH}(t)}{1 + c_{FSH,E}E_2(t)^2} \tag{3}$$

where $RP_{FSH}(t)$ is FSH produced in the pituitary awaiting release into the bloodstream, $FSH(t)$ is FSH blood concentration, $\theta_3$ is the circadian amplitude of FSH, and $\theta_4$ is the FSH circadian acrophase at time $t$ in days.

The estrogen (E$_2$) circadian rhythm was introduced by modifying its production terms from the original form

$$E_2(t) = e_0 + e_1 GrF(t) + e_2 DomF(t) + e_3 Lut_4(t) \tag{14A}$$

as follows:

$$E_2(t) = (e_0 + e_1 GrF(t) + e_2 DomF(t) + e_3 Lut_4(t))(1 + \theta_5 \cos(2\pi(t - \theta_6))) \tag{14B}$$

where $GrF(t)$ and $DomF(t)$ are active follicular masses in the growing and dominant stages, respectively, $Lut_4(t)$ is the active luteal mass in the fourth luteal stage, $\theta_5$ describes the circadian amplitude of E$_2$, and $\theta_6$ is the E$_2$ circadian acrophase at time $t$ in days.

Then, the progesterone (P$_4$) circadian rhythm was added by modifying its original production terms

$$P_4(t) = p_0 + p_1 Lut_3(t) + p_2 Lut_4(t) \tag{15A}$$

as follows:

$$P_4(t) = (p_0 + p_1 Lut_3(t) + p_2 Lut_4(t))(1 + \theta_7 \cos(2\pi(t - \theta_8))) \tag{15B}$$

where $Lut_3(t)$ is the active luteal mass in the third luteal stage, $\theta_7$ describes the circadian amplitude of P$_4$, and $\theta_8$ is the P$_4$ circadian acrophase at time $t$ in days.

**Incorporation of drug pharmacokinetics (PK) into the baseline model.** We then incorporated the PK of EE and DNG into the baseline model using a two-compartment model for extravascular (oral) administration (see S1 Text for details) [79]. This PK model, which showed a lower Akaike Information Criterion (AIC) than the one-compartment alternative (Fig B in S1 Text), was selected for its superior fit to the extracted EE and DNG drug concentration data (Fig 2C(i)) and is widely used in the literature [79]. It was then used to simulate the EE and DNG concentrations following a single administration. The PK model for EE and DNG is defined as follows:

$$C_{j,i}(t) = \begin{cases} 0, & \text{if } t < t_i + i - 1 \\ N_j e^{-k_{a,j}(t-(t_i+i-1))} + L_j e^{-\alpha_{1,j}(t-(t_i+i-1))} + M_j e^{-\beta_{1,j}(t-(t_i+i-1))}, & \text{if } t \geq t_i + i - 1, \end{cases}$$

where $N_j = \frac{k_{a,j}F_j u_{j,i}(k_{21,j}-k_{a,j})}{V_{c,j}(\alpha_{1,j}-k_{a,j})(\beta_{1,j}-k_{a,j})}$, $L_j = \frac{k_{a,j}F_j u_{j,i}(k_{21,j}-\alpha_{1,j})}{V_{c,j}(k_{a,j}-\alpha_{1,j})(\beta_{1,j}-\alpha_{1,j})}$, and $M_j = \frac{k_{a,j}F_j u_{j,i}(k_{21,j}-\beta_{1,j})}{V_{c,j}(k_{a,j}-\beta_{1,j})(\alpha_{1,j}-\beta_{1,j})}$ for $j \in \{EE, DNG\}$. Further, $C_{j,i}(t)$ is the drug concentration at time $t$ due to dose $u_{j,i}$, administered at time $t_i$ (ranging from 1:00–24:00) on day $i$ ($i = 1, \ldots, 28$). $F_j$ is the bioavailability, $k_{a,j}$ is the absorption rate constant, $k_{21,j}$ is the transfer rate from peripheral to central compartment, and $\alpha_{1,j}$, $\beta_{1,j}$ represent the distribution and elimination phases, respectively.

Next, we summed these individual EE and DNG dose concentrations to obtain exogenous estrogen ($E_2^{exo}$) and exogenous progesterone ($P_4^{exo}$) at time $t$. That is, $E_2^{exo}(t) = \sum_{i=1}^{28} C_{EE,i}(t)$ and $P_4^{exo}(t) = \sum_{i=1}^{28} C_{DNG,i}(t)$.

Then, because exogenous hormones do not interact identically with their endogenous counterparts we incorporated the RBA of EE and DNG. The original RBA values (1.9 for EE, 0.1 for DNG) were adjusted to 1.7 for EE and 0.01 for DNG to reflect the contraceptive effect of Valette (30 $\mu g$ EE and 2000 $\mu g$ DNG) [80]. These adjustments maintained the higher estrogen receptor affinity of EE relative to endogenous estrogen and the lower progesterone receptor affinity of DNG relative to endogenous progesterone. Then, to link exogenous and endogenous hormones, we introduced the coefficients $r_1$ (modified EE RBA value) and $r_2$ (modified DNG RBA value) and updated Eq 14B and Eq 15B as follows:

$$E_2(t) = (e_0 + e_1 GrF(t) + e_2 DomF(t) + e_3 Lut_4(t))(1 + \theta_5 \cos(2\pi(t - \theta_6)))$$
$$+ r_1 E_2^{\text{exo}}(t), \tag{14}$$

and

$$P_4(t) = (p_0 + p_1 Lut_3(t) + p_2 Lut_4(t))(1 + \theta_7 \cos(2\pi(t - \theta_8))) + r_2 P_4^{\text{exo}}(t). \tag{15}$$

## Parameter estimation

We estimated the new parameters describing circadian and PK components. Specifically, the circadian parameters of the new model were obtained through the following steps. First, we extracted the circadian rhythms of luteinizing hormone (LH), follicle-stimulating hormone (FSH), estrogen ($E_2$), and progesterone ($P_4$) during the follicular phase from Rahman et al. [20]. Second, we normalized the data by dividing their average values so that their averages become one (Fig 2B(i)). Third, we fitted the normalized data with a cosine curve $a\cos(2\pi(t - b)) + 1$, where $a$ represents the amplitude, $b$ is the acrophase, and $t$ denotes time in days. We then incorporated the fitted cosine curves into the simulated hormone levels from the baseline model (Fig 2B(ii)) to generate circadian rhythms of hormones throughout a 28-day cycle. That is, LH with circadian rhythm ($LH_{target}$) was obtained from the simulated hormone level from the baseline model ($LH$) by using $LH_{target}(t) = LH(t) + LH(t)(a\cos(2\pi(t - b)))$ (Fig 2B(iii), black). This ensures that higher hormone concentrations correspond to greater circadian oscillations [20]. This process was then repeated for the remaining hormones FSH, $E_2$, and $P_4$ to obtain their circadian rhythms ($FSH_{target}$, $E_{2,target}$, and $P_{4,target}$). Finally, to simulate these target circadian rhythms with our new model, we estimated the circadian parameters of the new model (Eqs 1, 3, 14, and 15). Specifically, we estimated the circadian parameters minimizing the least squares error between the target circadian rhythms and the simulated hormones with the new model (Fig 2B(iii)). The optimization was performed using MATLAB R2023b's *fminsearch* function (Nelder-Mead simplex algorithm) [81].

The PK parameters of the drug model were estimated using the extracted EE and DNG plasma concentration data (Fig 2C(i)). These parameters for absorption rate ($k_{a,j}$), transfer rate ($k_{21,j}$), distribution rate ($\alpha_{1,j}$), elimination rate ($\beta_{1,j}$), and volume of distribution ($V_{c,j}$) for drug component $j \in \{EE, DNG\}$ were estimated by minimizing the residual sum of squares between simulated and experimental concentrations (Fig 2C(iii)). This optimization was also performed using the *fminsearch* function in MATLAB.

## Optimal regimen for nonconstant drug administration

Using the estimated PK parameters for our drug model, we explored optimal dosing regimens across different dosing times for both constant and nonconstant administration. To determine the optimal nonconstant dosing regimen, we allowed daily doses of EE and DNG to vary between 0 and the market doses (30 µg EE, 2000 µg DNG) over 21 days, followed by 7 drug-free days. Then, we optimized the daily EE and DNG doses for each dosing time ranging from 1:00–24:00.

We employed a multi-objective optimization approach, which is particularly useful when multiple competing objectives must be balanced [82]. In this case, there is a trade-off between reducing the maximum $P_4$ level to ensure anovulation and minimizing the total EE and DNG doses to reduce drug exposure. We aimed to meet these two competing objectives, which were defined as follows: maximum $P_4$ level over three cycles (84 days) and total 21-day EE and DNG doses. We performed optimization over three cycles instead of ten as a practical compromise to reduce computational burden. This duration was chosen based on Spona et al. [5], who indicated that three cycles can

provide a reasonable basis for assessing contraceptive efficacy. The mathematical formulation of the problem is as follows:

$$\min_{u_{EE}, u_{DNG}} J(u_{EE}, u_{DNG}) = \begin{bmatrix} J_1(u_{EE}, u_{DNG}) \\ J_2(u_{EE}, u_{DNG}) \end{bmatrix}$$

subject to the mathematical model $x'(t) = f(t, x(t), Inh(t - \tau), u_{EE}, u_{DNG})$ with initial conditions $x(t_0) = x_0$ (see S1 Text for details). Here, $x(t)$ denotes the state variables and the objective functions are defined as

$$J_1(u_{EE}, u_{DNG}) = \max_{t \in [0, 3 \text{ cycles}]} P_4(t)$$
$$J_2(u_{EE}, u_{DNG}) = \sum_{i=1}^{21} u_{EE,i} + u_{DNG,i},$$

where the control variables are $u_{EE} = \{u_{EE,1}, \ldots, u_{EE,21}\}$ and $u_{DNG} = \{u_{DNG,1}, \ldots, u_{DNG,21}\}$, with $u_{EE,i}$ and $u_{DNG,i}$ representing EE and DNG doses for day $i$, which satisfy $0 \leq u_{EE,i} \leq 30$ $\mu$g and $0 \leq u_{DNG,i} \leq 2000$ $\mu$g for $1 \leq i \leq 21$.

To solve this multi-objective control problem, we employed the Julia package Metaheuristics.jl v3.3.5 [83], which implements the CCMO (Coevolutionary Constrained Multi-Objective Optimization) framework with NSGA-II (Non-dominated Sorting Genetic Algorithm II) as its optimizer [83–85]. NSGA-II is an evolutionary algorithm according to genetic algorithm (GA), a randomized, population-based search strategy based on principles of genetics. GA begins by generating an initial population of candidate solutions, evaluating them based on their objective functions. Then, new populations with an average of objective function value that is lower than that of the preceding population are iteratively generated through crossover and mutation operations, ensuring progressive improvement in solution quality (see [82] for more details). The process continues until the function tolerance criterion (set to $10^{-4}$ here) is met. The CCMO framework enhances this process by utilizing a dual-population setup (set to population sizes of 400 and 500), ensuring robust exploration of the solution space.

Next, a confirmation step was applied: from the generated solutions, we selected the regimen with the lowest total EE and DNG doses that maintained anovulatory $P_4$ levels over 10 cycles, demonstrating long-term contraceptive efficacy. Yet the dosing regimens included very low doses that could present formulation challenges such as blend uniformity, dose precision, and product stability [86,87]. We found that setting these low doses (i.e., below 1 µg) to zero, did not affect ovulation suppression and yielded a more optimal dosing scheme due to the reduced total dose. Accordingly, the modified dosing regimen is temporarily designated as the optimal scheme for that specific dosing time. Then, because metaheuristic optimization can yield locally optimal rather than truly minimal doses, we added a validation step to confirm that the final minimum dose for each dosing time was the lowest effective dose across all optimization results. First, the lowest-dose combination from the temporary optimal schemes was tested at every dosing time (01:00–24:00). If anovulation occurred at any time point, that dose–time pair was recorded as the optimal regimen for those times. The next lowest-dose combination was then tested on the remaining dosing times, and any effective regimens were similarly recorded. This process continued iteratively until all dosing times were evaluated and no further dose reductions were possible without compromising contraceptive efficacy.

## Supporting information

**S1 Text. Model details and supplementary figures.**
(PDF)

## Acknowledgments

We thank the members of the Biomedical Mathematics Group at the Institute for Basic Science for their insightful discussions, Prof. Aurelio A. de los Reyes V for his contributions to the development of the model and the formulation of the optimal control approach, and the Computational Research Laboratory of the Institute of Mathematics at UP Diliman for providing facilities where some simulations were generated.

## Author contributions

**Conceptualization:** Jae Kyoung Kim.

**Data curation:** Brenda Lyn Gavina, Taeyong Lee, Soyoung Kim, Jae Kyoung Kim.

**Formal analysis:** Brenda Lyn Gavina, Taeyong Lee, Olive Cawiding, Sunhwa Choi, Sungwook Choi, Soyoung Kim, Jae Kyoung Kim.

**Funding acquisition:** Sunhwa Choi, Soyoung Kim, Jae Kyoung Kim.

**Investigation:** Brenda Lyn Gavina, Taeyong Lee, Olive Cawiding, Sunhwa Choi, Sungwook Choi, Soyoung Kim, Jae Kyoung Kim.

**Methodology:** Brenda Lyn Gavina, Taeyong Lee, Soyoung Kim, Jae Kyoung Kim.

**Project administration:** Jae Kyoung Kim.

**Resources:** Jae Kyoung Kim.

**Software:** Brenda Lyn Gavina, Taeyong Lee, Olive Cawiding, Soyoung Kim, Jae Kyoung Kim.

**Supervision:** Soyoung Kim, Jae Kyoung Kim.

**Validation:** Brenda Lyn Gavina, Taeyong Lee, Soyoung Kim, Jae Kyoung Kim.

**Visualization:** Brenda Lyn Gavina, Taeyong Lee, Olive Cawiding, Soyoung Kim, Jae Kyoung Kim.

**Writing – original draft:** Brenda Lyn Gavina, Taeyong Lee, Olive Cawiding, Sunhwa Choi, Sungwook Choi, Soyoung Kim, Jae Kyoung Kim.

**Writing – review & editing:** Brenda Lyn Gavina, Taeyong Lee, Olive Cawiding, Sunhwa Choi, Sungwook Choi, Soyoung Kim, Jae Kyoung Kim.

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
