## [Decision Letter · Decision Letter 0]

22 May 2025

Optimizing oral contraceptive timing: Morning intake reduces doses and enhances efficacy

PLOS Computational Biology

Dear Dr. Kim,

Thank you for submitting your manuscript to PLOS Computational Biology. After careful consideration, we feel that it has merit but does not fully meet PLOS Computational Biology's publication criteria as it currently stands. Therefore, we invite you to submit a revised version of the manuscript that addresses the points raised during the review process.

Please submit your revised manuscript within 60 days Jul 22 2025 11:59PM. If you will need more time than this to complete your revisions, please reply to this message or contact the journal office at ploscompbiol@plos.org. Please include the following items when submitting your revised manuscript:

We look forward to receiving your revised manuscript.

Kind regards,

Marc Birtwistle

Section Editor

PLOS Computational Biology

**Journal Requirements:**

At this stage, the following Authors/Authors require contributions: Jae Kyoung Kim. Please ensure that the full contributions of each author are acknowledged in the "Add/Edit/Remove Authors" section of our submission form.

Potential Copyright Issues:

i) Figures 1, 3, 4A, 4C, 5A, 5D, S1 Fig2(A, and B). Please confirm whether you drew the images / clip-art within the figure panels by hand. If you did not draw the images, please provide (a) a link to the source of the images or icons and their license / terms of use; or (b) written permission from the copyright holder to publish the images or icons under our CC BY 4.0 license. Alternatively, you may replace the images with open source alternatives. See these open source resources you may use to replace images / clip-art:

**Reviewers' comments:**

Reviewer's Responses to Questions

**Comments to the Authors:**

**Please note that one of the reviews is uploaded as an attachment.**

Reviewer #1: The authors introduce circadian rhythms into a previously published mathematical model of the female menstrual cycle to analyze the impact of daytime of oral contraceptive (OC) dosing on the required dose to suppress ovulation. Their model predicts that the required dose can be reduced significantly with morning intake compared to evening intake, and that further reduction is possible with non-constant dosing regimens including intake-free days.

The article is well written and nicely illustrates the potential use of mechanistic models for dose optimization. However, the manuscript ignores important aspects of OCs which need to be addressed to make this work practically relevant instead of a sole computational exercise.

1. The results seem to be highly model- and parameter-specific. While testing the predictions with other models might go beyond the scope of the manuscript, testing other parameter regimens is possible (please also see my 2nd comment below). How robust are the findings w.r.t. to changes in model parameters?

2. The menstrual cycle is characterized by a high within- and between-subject variability, which is not accounted for in the model. The results seem to be valid only for a regular 28-day cycle, which occurs rarely. The authors should test other parametrizations of the model that result in different cycle lengths or quasi-periodic cycles (e.g. as in Hendrix et al., Bull Math Biol (2014) 76:136–156

DOI 10.1007/s11538-013-9913-7).

3. There exists a wide variety of oral contraceptives that differ in the PK and PD profiles, but the authors selected only one, arguing that it is particularly susceptible to circadian variation. What about other OCs with longer half-lifes? Would the findings be different?

4. Is the maintenance of P4 periodicity under constant OC intake a model artifact or are there data showing this? I believe it is an artifact as there should not be such a pattern if no corpus luteum is formed.

5. The data by Welt have only a resolution on the level of days, but do not contain information about daytime, meaning that the simulated hormone curves without circadian rhythms could be shifted by +/- 12 hours. How does such shifting impact the results?

6. The fit to the exogenous hormone data looks rather poor. Have the authors considered using a two-compartment model?

Minor comments:

7. What was the stepsize for testing different doses in a certain range. Was it steps of 1 microgram?

8. S1 Table 1: The units need to be formatted properly.

Reviewer #2: Please see attached.

**Have the authors made all data and (if applicable) computational code underlying the findings in their manuscript fully available?**

Reviewer #1: Yes

Reviewer #2: **No:** One cannot replicate the figures with what is presented.

PLOS authors have the option to publish the peer review history of their article (what does this mean? ). If published, this will include your full peer review and any attached files.

**Do you want your identity to be public for this peer review?** For information about this choice, including consent withdrawal, please see our Privacy Policy .

Reviewer #1: No

Reviewer #2: No

**Figure resubmission:**
---

## [Decision Letter · Decision Letter 1]

29 Oct 2025

PCOMPBIOL-D-25-00553R1

Optimizing oral contraceptive timing: Daytime intake reduces doses and enhances efficacy

PLOS Computational Biology

Dear Dr. Kim,

Thank you for submitting your manuscript to PLOS Computational Biology. After careful consideration, we feel that it has merit but does not fully meet PLOS Computational Biology's publication criteria as it currently stands. Therefore, we invite you to submit a revised version of the manuscript that addresses the points raised during the review process.

Please submit your revised manuscript within 30 days Dec 29 2025 11:59PM. If you will need more time than this to complete your revisions, please reply to this message or contact the journal office at ploscompbiol@plos.org. Please include the following items when submitting your revised manuscript:

We look forward to receiving your revised manuscript.

Kind regards,

Marc Birtwistle

Section Editor

PLOS Computational Biology

**Reviewers' comments:**

Reviewer's Responses to Questions

**Comments to the Authors:**

Reviewer #1: The authors have done a huge effort in addressing all my comments and questions. They have substantiated their findings by performing extensive parameter studies and provided clarification on technical details. I have no further comments or suggestions for improvement of the manuscript, despite one last question. I wonder if the authors can provide an explanation for why the total required DNG dose (and partially also the EE dose) is lower in the +12h shift scenario (Fig. 4B in S1) compared to the -12h shift. Is it also lower compared to the original (no shift) scenario? They should briefly comment on this in the results section (page 7, line 171).

Reviewer #2: Thanks to the authors for nicely replying to my comments and questions. A few small points remain to be addressed as below.

-First of all, the code online looks much better and can replicate the figure panels. However, there are still quite a few hard-coded lines that are not well explained. For instance, lines 115-136 in Fig2C_iii.m are not at all intuitive on what is going on.

-The new two-compartment model needs to be explained a bit more. What are the compartments, the reaction formalisms, reasons, the math.

-Finally, I am still skeptic on the choice of 3 ng/mL progesterone as the criteria used to define anovulation. What if 5 was chosen here as well? It could be discussed and showed simulation-wise that it is okay to deviate as such from the base Gavina model used.

**Have the authors made all data and (if applicable) computational code underlying the findings in their manuscript fully available?**

Reviewer #1: Yes

Reviewer #2: Yes

PLOS authors have the option to publish the peer review history of their article (what does this mean? ). If published, this will include your full peer review and any attached files.

**Do you want your identity to be public for this peer review?** For information about this choice, including consent withdrawal, please see our Privacy Policy .

Reviewer #1: No

Reviewer #2: No

**Figure resubmission:**
---

## [Decision Letter · Decision Letter 2]

14 Jan 2026

PCOMPBIOL-D-25-00553R2

Optimizing oral contraceptive timing: Daytime intake reduces doses and enhances efficacy

PLOS Computational Biology

Dear Dr. Kim,

Thank you for submitting your manuscript to PLOS Computational Biology. After careful consideration, we feel that it has merit but does not fully meet PLOS Computational Biology's publication criteria as it currently stands. Therefore, we invite you to submit a revised version of the manuscript that addresses the points raised during the review process.

We look forward to receiving your revised manuscript.

Kind regards,

Marc Birtwistle

Section Editor

PLOS Computational Biology

**Additional Editor Comments:**

Please do give careful consideration to the github repository and its ability to enable sharing and reuse of this work.

**Reviewers' comments:**

Reviewer's Responses to Questions

**Comments to the Authors:**

Reviewer #1: The authors have addressed all my comments and I recommend acceptance of the manuscript.

Reviewer #2: I thank the authors for the rebuttal. However, I feel that my comments are not satisfactorily answered. Mainly, the GitHub repository code still lacks enough comments and contains too many hard-coded numbers for plotting/simulating. The repo needs a thorough rework to comply with reproducibility. Secondly, the increase of anovulation threshold seemed to cause losing the DNG levels in the provided Fig.R. Why is that?

**Have the authors made all data and (if applicable) computational code underlying the findings in their manuscript fully available?**

Reviewer #1: Yes

Reviewer #2: Yes

PLOS authors have the option to publish the peer review history of their article (what does this mean? ). If published, this will include your full peer review and any attached files.

**Do you want your identity to be public for this peer review?** For information about this choice, including consent withdrawal, please see our Privacy Policy .

Reviewer #1: No

Reviewer #2: No

**Figure resubmission:**
---

## [Decision Letter · Decision Letter 3]

18 Feb 2026

Dear Professor Kim,

We are pleased to inform you that your manuscript 'Optimizing oral contraceptive timing: Daytime intake reduces doses and enhances efficacy' has been provisionally accepted for publication in PLOS Computational Biology.

Best regards,

Marc Birtwistle

Section Editor

PLOS Computational Biology

Reviewer's Responses to Questions

**Comments to the Authors:**

Reviewer #2: Thank you to the authors for enhancing the repository and for their answer.

**Have the authors made all data and (if applicable) computational code underlying the findings in their manuscript fully available?**

Reviewer #2: Yes

PLOS authors have the option to publish the peer review history of their article (what does this mean? ). If published, this will include your full peer review and any attached files.

**Do you want your identity to be public for this peer review?** For information about this choice, including consent withdrawal, please see our Privacy Policy .

Reviewer #2: No

---

## [Editor Report · Acceptance letter]

PCOMPBIOL-D-25-00553R3

Optimizing oral contraceptive timing: Daytime intake reduces doses and enhances efficacy

Dear Dr Kim,

I am pleased to inform you that your manuscript has been formally accepted for publication in PLOS Computational Biology. Your manuscript is now with our production department and you will be notified of the publication date in due course.

With kind regards,

Zsofia Freund
